# A Detailed Liquefaction Susceptibility Map of Nestos River Delta, Thrace, Greece Based on Surficial Geology and Geomorphology

Maria Taftsoglou [1], Sotirios Valkaniotis [1], George Papathanassiou [2,*], Nikos Klimis [1] and Ioannis Dokas [1]

1 Department of Civil Engineering, Democritus University of Thrace, University Campus, 671 00 Xanthi, Greece
2 Department of Geology, Aristotle University of Thessaloniki, 541 24 Thessaloniki, Greece
* Correspondence: gpapatha@geo.auth.gr

**Abstract:** The existence of high potential onshore and offshore active faults capable to trigger large earthquakes in the broader area of Thrace, Greece in correlation with the critical infrastructures constructed on the recent and Holocene sediments of Nestos river delta plain, was the motivation for this research. The goal of this study is twofold; compilation of a new geomorphological map of the study area and the assessment of the liquefaction susceptibility of the surficial geological units. Liquefaction susceptibility at regional scale is assessed by taking into account information dealing with the depositional environment and age of the surficial geological units. In our case, available geological mapping shows a deficient depiction of Pleistocene and Holocene deposits. Taking into consideration the heterogeneously behavior of active floodplains and deltas in terms of liquefaction, a detailed classification of geological units was mandatory. Using data provided by satellite and aerial imagery, and topographic maps, dated before the 1970's when extensive modifications and land reclamation occurred in the area, we were able to trace fluvial and coastal geomorphological features like abandoned stream/meanders, estuaries, dunes, lagoons and ox-bow lakes. This geomorphological-oriented approach clearly classified the geological units according to their depositional environment and resulted in a more reliable liquefaction susceptibility map of 4 classes of susceptibility; Low, Moderate, High and Very High. The sediments classified as very high liquefaction susceptibility are related to fluvial landforms, the high to moderate liquefaction susceptibility ones in coastal and floodplain landforms, and low susceptibility in zones of marshes. The sediments classified in the highest group of liquefaction susceptibility cover 85.56 km$^2$ of the study area (16.6%). Particular attention was drawn to critical infrastructure (Kavala International Airport "Alexander the Great") constructed on the most prone to liquefaction areas.

**Keywords:** liquefaction; susceptibility; Thrace; remote sensing; geomorphology

## 1. Introduction

Liquefaction is a natural process that can be triggered by earthquakes in saturated loose sandy soils covered with impermeable sediments within a certain distance from the epicenter of an earthquake. The term 'liquefaction' was originally introduced by Mogami and Kubo in 1953 [1]. In particular, when a seismic event shakes the ground, increase in pore water pressure and reduction in effective stresses occurs, resulting in the transformation of the unconsolidated and saturated granular soil from solid state to a liquid one [2]. Because of the peculiar subsurface stratigraphy, this is a common occurrence in alluvial and coastal plains [3].

Despite the fact that liquefaction is considered less hazardous than other secondary effects induced by earthquakes, e.g., landslides, significant failures in infrastructure, agricultural lands and properties have been recently reported in the cases of the 2010–2011 Canterbury earthquake sequence in New Zealand [4], the 2012 Emilia earthquake sequence in the Po Plain, northern Italy [5], and the 2014 Cephalonia earthquake sequence in

Greece [6]. Regarding the 20th Century, severe liquefaction-related damages were induced by the 1964 M9.2 Alaska earthquake [7] and the M7.5 Niigata earthquake [8], as well as by the 1995 M6.9 Kobe (Japan), the 1999 M7.5 Chi-Chi (Taiwan) and the 1999 M7.4 Izmit (Turkey) earthquakes [9–12].

It is evident that information about the severity of ground shaking in comparison with the liquefaction susceptibility of each key study can contribute significantly to seismic hazard assessment at the local and regional scale [13–15]. In fact, a liquefaction susceptibility map can be used as a screening guide leading to hazard and risk maps for land-use planning purposes avoiding in advance prone to liquefaction areas.

Nowadays, liquefaction susceptibility assessment in fluvial and coastal plains has a growing interest because the fast growth of global population increases the need for new unoccupied areas for urbanization. Liquefaction susceptibility assessment is a procedure dealing with the physical properties of soil and the depth of water table, while considering the seismic factor in this equation; then, the liquefaction potential can be evaluated. From an engineering geological and geotechnical point of view, the most applied approaches for the estimation of the liquefaction potential of sediments are the Liquefaction Potential Index (LPI) and the Liquefaction Severity Index (LSN) [16–24].

In case of a regional study, the qualitative methods for assessing the liquefaction susceptibility are based on the correlation of the depositional environment and the age of the sediments. The most applied approaches are the ones proposed by Youd and Perkins [25], Wakamatsu [26], the California Department of Conservation Division of Mines and Geology [27] and Witter et al. [28].

Recently, the geomorphological studies carried out during the last decade in New Zealand, Japan, Italy and Greece concluded that variations in river morphology and associated depositional settings of sediments influence the observed manifestations of the liquefaction phenomena [29–34]. Specifically, focusing on the distribution of the liquefaction features, it is clearly shown that they are not randomly distributed over the floodplain areas but are mostly concentrated in clusters and rectilinear or meander-like alignments [35]. As shown in Canterbury, New Zealand, most liquefaction induced occurrences are related to paleo-channels, current river channels and point-bars [31]. In particular, according to [36], the typically younger and less consolidated sediments of inner meanders can generate phenomena of lateral spreading during a strong horizontal ground motion because they are not confined, in contrast with the outer confined part of the meanders. A recent study regarding 120 liquefied sites induced by the 2012 Emilia Romagna earthquakes reported that ejected granular sediments are entirely derived from Holocene units associated with river channel, levee and crevasse deposits [37]. Moreover, in the case of the March 2021 seismic sequence in Thessaly (Greece), liquefaction phenomena were reported not only on river channels and point-bars but also in abandoned channels and oxbow lakes, demonstrating the strong correlation of liquefaction manifestations with the depositional environment and the impact of their spatial distribution to the future assessment of the liquefaction susceptibility of the areas [33]. In addition, after the Durres earthquake (Albania), sand boils and water and sand fountains in Rinia-Fllake Lagoon were observed, strengthening the opinion that a depositional environment with shallow water and soft and unconsolidated nature of sediments are contributing factors for liquefaction phenomena [38].

In order to delineate the geomorphological features in the areas of interest, many researchers used historical maps and aerial and satellite imagery. In particular, historical data allow for the mapping of geomorphological features before the extensive modification of the areas by contemporary human activities. Due to this fact, areas prone to liquefaction processes, such as old-abandoned meanders, point-bars and coastal barriers that are more prone, can be traced in detail. Furthermore, aerial and satellite imagery can provide information immediately after an event contributing to the delineation of liquefaction features, even in inaccessible areas.

This study focuses on the floodplain of the Nestos River in Thrace, Greece. Thrace is generally considered a low seismicity zone according to historical and recent earthquakes in Greece [39]. However, tectonic structures capable of triggering big events and active faults are present in the broader area. Due to the current N–S extensional stress regime of the region, the general orientation of the active faults is W–E [40]. Most of the on-shore faults are normal–oblique, while off-shore major strike-slip faults along the North Aegean Trough, the westward continuation of the North Anatolian Fault Zone, are present. The major fault zones close to the Nestos Delta are the Kavala–Strymonas, Xanthi–Komotini, Maronia–Makri, Drama and the Orestias–Mikri Doxipara fault zones [41,42]. The first two seem to be connected, creating the extended Kavala–Xanthi–Komotini (Figure 1) active fault zone with a length of more than 100 km, which subdivides the regional unit into two different landscape parts, the northern mountainous part and the southern flat or plain part. In addition, large earthquakes can be produced by the western part of the North Anatolian Fault, the tectonic structure of the Saros Fault zone along the North Aegean Trough. Furthermore, critical infrastructures such as gas pipelines, airports and ports were developed over the Thrace floodplains and deltas; most of them on unconsolidated sediments. This regime, in conjunction to the recent and late Holocene deposits covering mostly estuaries of the Nestos and Evros rivers, were the main factors that motivated us to study the Nestos River plain and delta regarding the liquefaction susceptibility of its deposits.

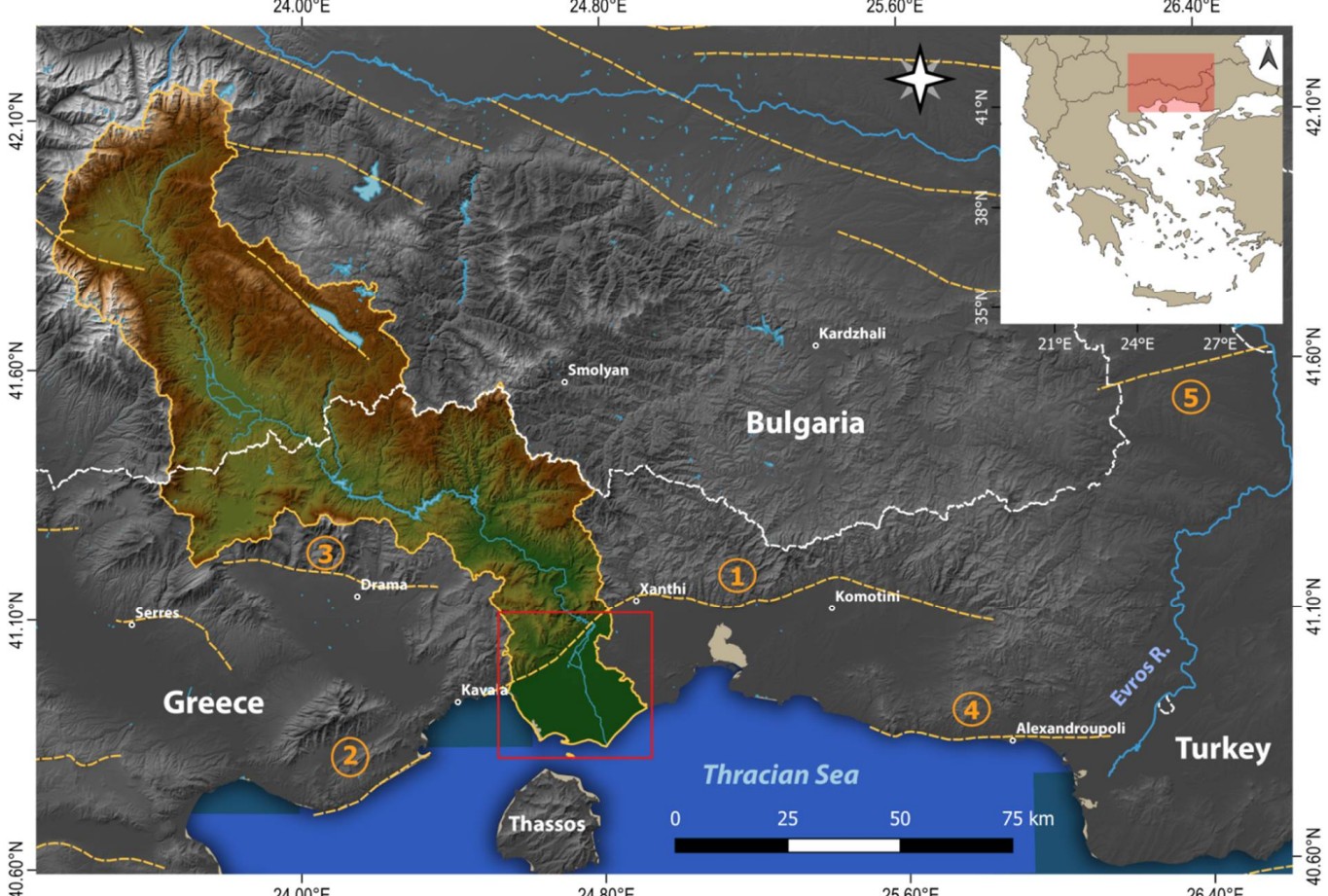

**Figure 1.** Geophysical map highlighting the Nestos River zone. The study area is indicated with the red square. (1) Kavala–Xanthi–Komotini fault, (2) Simvolo fault, (3) Drama fault, (4) Maronia fault, (5) Orestias–Mikri Doxipara. Fault zones from [41,42].

The goal of this study is to compile a liquefaction susceptibility map of the Nestos River delta based on the geomorphological evolution of the river. In order to achieve this, the geological map of the area (1:50,000) developed by HSGME (Hellenic Survey of Geology and Mineral Exploration) was primarily used as the base layer. In order to delineate the geomorphological features of Nestos, we used satellite and aerial images and topographic maps. This layer of information is focused on the evolution of river data and contributed to a significant improvement regarding the tracing and mapping of fluvial and coastal geomorphological features, such as abandoned stream/meanders, estuaries, dunes, lagoons and ox-bow lakes. As a result, surficial sediments were classified into liquefaction susceptibility classes according to their possible age and depositional environment.

*Geology-Geomorphology*

The regional unit of Thrace consists of formations attributed to the Rhodope mass and the Circum–Rhodope belt. In particular, the Rhodope metamorphic province comprises metamorphic and igneous rocks of thickness about 24 km [43]. The crystalline–schist bedrock of Rhodope Massif is detected in the mountainous massif northwards of Komotini and Alexandroupolis towns and is divided in two tectonic units, Pangaio and Sidironero, which thrust over Pangaion from the north to the south along a tectonic line of NW–SE direction that ends in E–W [43]. The Circum–Rhodope Belt (CRB) fringing the Rhodope Massif in northern Thrace consists of phyllites, green schists and post-sedimentary rocks with intercalations of igneous rocks. This zone is overthrust on the Rhodope mass and continues upwards with semi-metamorphic formations [44].

Igneous rocks like granites, granodiorites, monzonites and diorites of Eocene-Miocene are distributed mostly on south east of Thrace and north of Xanthi, along the Greece-Bulgarian borderline. Molassic series of conglomerates, sandstones, marls and marly limestones with lignite horizons dates from Middle Eocene-Oligocene and probably up to the base of Miocene [45].

Lowland areas of Thrace are covered by Neogene and Quaternary deposits of sands, gravels, cobbles and clays extending in the Evros Delta and alluvial deposits in the areas of the Nestos delta and in the center of the Xanthi–Komotini basin. The study area of the Nestos delta is a part of the Xanthi–Komotini post-alpine basin extended from the Rhodope mountainous area to the sea in a huge fan-shaped form (Figures 1 and 2). The bedrock of the basin consists of crystalline-schistosed rocks, such as gneisses, schists, aphivolites and thick marble layers of the Rhodope Massif. Boreholes on both the terrestrial deltaic zone of Nestos and the surrounding continental shelf zone [45–49] have provided information for geomorphological deposits of the area since the Neogene period. In particular, the initial subsidence and faulting of the basin started in the Lower to Middle Miocene in the rift area between Thassos island and Kavala Bay [46,48]. During the Miocene, marine and coastal sediments were deposited consisting of white porous limestones, sandstones, marls and clays, as well as lacustrine deposits, consisting of alternations of sandstones, sands and sandy clays. The coastal sediments were formed after a period of intense evaporation (Upper Miocene). In the Pliocene period, yellowish, sandy marls, loams and clays with layers of marly limestones were deposited. Deposition of clastic Pleistocene deposits occurred in a coastal–deltaic environment [49] covering unconformably Miocene units. Pleistocene sediments consist of talus cones, scree, reddish clays and sands with lenses of non-cohesive conglomerates. Gravels, cobbles and silts of the Pleistocene age are also met on the surface of foothills as a result of the fluvial processes and erosion in the upper limits of the plain. Quaternary and recent deposits comprise alternations of silts, sands, clays and gravels deposited principally in a deltaic environment with a thickness of 100–200 m. Generally, the continuous subsidence of the basin resulted in thick sedimentation with a range approximately from 2.5 to 6.0 km [50].

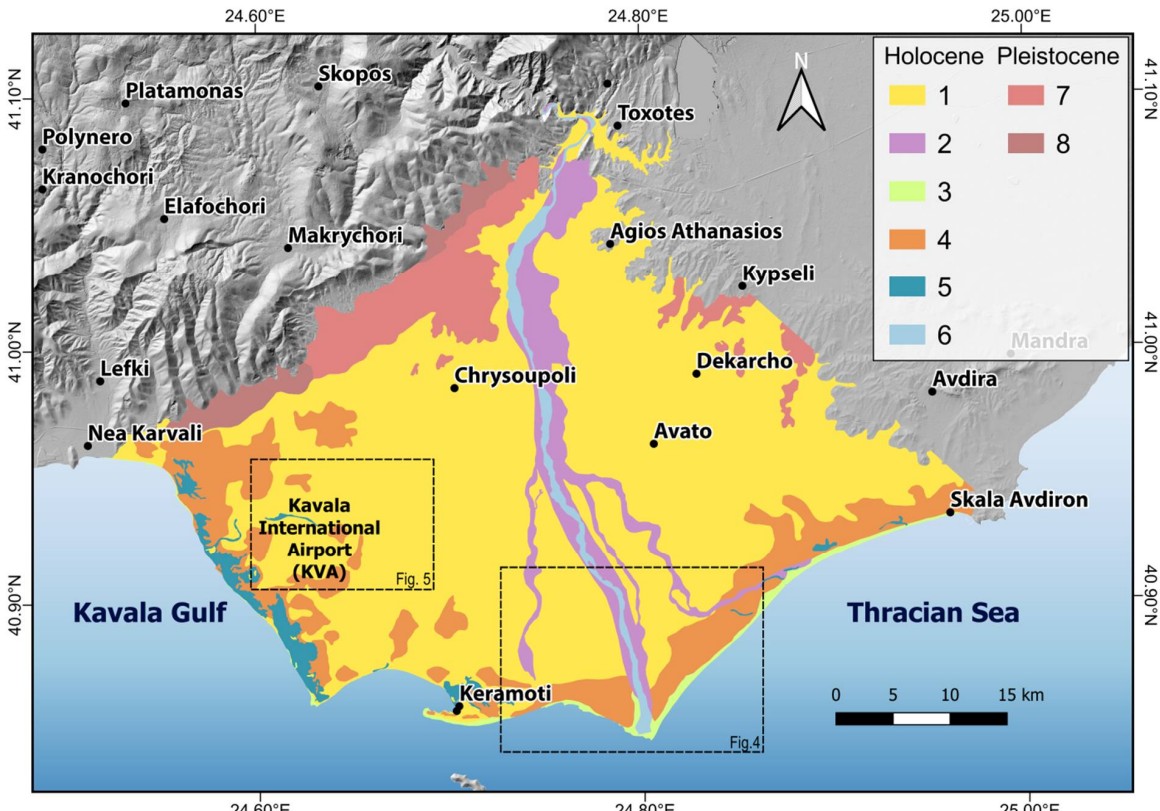

**Figure 2.** Geological map of the Nestos River delta provided by HSGME with (1) floodplain deposits, (2) channel deposits, (3) coastal deposits, (4) swamp deposits, (5) lagoons, (6) Nestos River, (7) Pleistocene deposits, (8) screes.

## 2. Materials and Methods

The depositional processes affect the liquefaction susceptibility of sediments since fine and coarse grained soils sorted by fluvial or wave actions are more susceptible than unsorted sediments [51]. Collecting data regarding the depositional environments, the age of deposits and the groundwater depth, the liquefaction susceptibility can be defined in a regional scale.

In particular, one of the most applied qualitative methods is based on the Youd and Perkins criteria [25]. According to this approach, the geological units are classified in five classes of liquefaction susceptibility (very low, low, moderate, high and very high), based on the age of the sediments, the depositional process and the depth of the water table in the area of interest. As a result, younger, looser and more segregated deposits are classified as more prone to the liquefaction phenomena.

The susceptibility to liquefaction of the surficial deposits within the Nestos River plain was assessed by applying the Youd and Perkins criteria [25]. In order to achieve this, it was important to define the age and depositional environment of the deposits and the depth of water. The latter one, depth of ground water, was evaluated based on a regional borehole dataset compiled by hydrological surveys [49,52–54]. The only available delta-wide groundwater survey dates from 1978 and is presented in Figure S1 [49,55]. Due to the lack of more recent regional surveys, we estimate the current groundwater levels based on the 1978 map and the 2014 survey results in the eastern part of the Nestos delta by [54]. The piezometric surface of the phreatic/unconfined aquifer has risen in recent years, probably due to the abandonment of shallow boreholes and adoption of surface irrigation and deep boreholes [54]. The depth of groundwater in areas of the Holocene, alluvial, fluvial and coastal sediments were evaluated as less than 6 m, taking into consideration the seasonal saturation of unsaturated soils and variation of the phreatic/unconfined aquifer.

In order to evaluate the age and the depositional environment of the geological units, we had to initially consider the geological map of the area. In addition, we used supplementary data (aerial and satellite imagery) to produce a geomorphological map where detailed information would be defined.

### 2.1. Geological Map of Nestos River Delta

The geological maps in 1:50,000 scale, developed by HSGME in four separate map sheets, were used as the basic data layers for the initial depiction of the area of interest [56–59]. According to these maps, the area of the Nestos River plain consists of eight geological units. As Holocene deposits are described fluvial sediments of old-abandoned beds and terraces of Nestos, which are situated parallel to the river and consist of alluvial deposits of clays, sands and gravels while floodplain deposits consist mainly of clays and sands extended in the whole area. Furthermore, marsh deposits of organic clays and silts were mostly found in the west part of the plain parallel to the coast, where coastal deposits with loose sands and locally pebbles and gravels were additionally mapped. In the NW region of the plain, Pleistocene sediments of screes and talus cones were also detected, mainly on the foothills, originating from fluvial processes and erosion of the upper limits of the plain. In the middle of this area, the current main river channel of Nestos is forming while lagoon formations parallel to the shoreline are depicted (Figure 2).

### 2.2. Historical Orthophoto Maps

The first aerial photography coverage of Greece was acquired by Royal Air Force in 1945. It lacks fiducial marks and camera calibration data, because it was taken by reconnaissance cameras. Historical aerial photographs of 1945, and diapositive transparency reproductions were obtained from Hellenic Mapping and Cadastral Organization (HEMCO) through license of the Hellenic Military Geographical Service (HMGS), which maintains a repository of 13,200 negatives, in the form of individual frames. Diapositives were scanned at 1700 dpi at 8-bit grayscale using high quality photogrammetric scanners at the contractors' sites. Orthophoto image accuracy is reported in ground distance and is less than ±2 m. A unified orthomosaic of continental Greece for 1945 is available by HMGS and Hellenic Cadastre.

In order to collect more information for the geomorphological formations of Nestos river plain, it was important to know the environmental status before the influence of the anthropogenic factor. Thus, orthophoto maps of 1945 provided by Hellenic Cadastre organization were used in our study as the basic historical imagery layer (ground resolution 2 m), avoiding extensive modifications, land reclamation and irrigation crop development during the second half of 20th century. We also use Hellenic Cadaster orthophotos of 2007-9 as a reference layer for modern setting of the area.

### 2.3. Declassified Satellite Imagery

The US Corona reconnaissance program operated between 1959 and 1972 as part of the US Key Hole program and consisted of a series of low-Earth orbit satellite missions.

The KH-4 missions started in 1962 as the first satellites acquiring stereo imagery from space. They consisted of two panoramic cameras; the fore (forward) and aft (backward); tilted respectively 15° at an orbit of ~200 km. The cameras are characterized by a focal length of 602.6 mm, a fixed aperture width of 5.265°, a pan angle of 71.16° and a light sensitive panchromatic film with a resolution between 50 and 160 lines/mm.

Comparable with the orthophoto maps, declassified images were used in order to analyze the geomorphological evolution of the Nestos River plain during a period of a more intense intervention of anthropogenic factor. In this study, we used KH-4 Corona frames (Table 1) with a 2–4 m ground resolution, acquired in August 1960 and August 1968 from USGS/NARA. KH-4 frames were orthorectified in ERDAS IMAGINE 2020 using ground control points picked from the 2007–2009 orthophotos and DEM in order to resolve frame intrinsic and distortions. Ground control points were added until sufficient

residual values were reached in triangulation and the final orthophoto maps were visually checked by overlaying a street vector layer from OpenStreetMap (Tables S1–S3, Figure S2 in Supplementary Materials).

**Table 1.** Specifications of aerial and satellite imagery used in this study.

| Data | Date | Spatial Resolution | Source |
|---|---|---|---|
| Historical Orthophoto maps | 1945 | 2 m | Hellenic Cadastre–HMGS |
| Orthophoto maps | 2007–2009 | 0.5 m | Hellenic Cadastre |
| Corona KH-4 | 18 August 1960<br>11 August 1968 | 2–4 m<br>2–4 m | USGS/NARA |
| DEM | - | 5 m | Hellenic Cadastre |

*2.4. Methods*

Applying the criteria proposed by [25], we initially searched for information on the geological maps in a 1:50,000 scale. However, we observed that this layer lacks geomorphological details, especially in river and coastal formations. In most of the areas, these official published geological maps showed undivided alluvial and floodplain deposits. In particular, only the current part of the Nestos River channel was depicted with the fluvial deposits detected only around it. A similar lack of detail was also observed in deltaic deposits, which together with coastal formations were not identified.

For overcoming this lack of information, we used orthophoto maps from 1945 derived from Hellenic Cadastre (Figures 3–5). We observed that the area is covered by an extended hydrographic network, where different geomorphological features of river could be traced. In particular, darker areas through the river network depict an earlier generation of paleochannels inferred to be old meanders of the Nestos River and point-bars associated with the abandoned meanders. Furthermore, dark spots around the floodplain were specified as swamp areas and brighter ones near the coast as deltaic. Parallel to the shoreline of the orthophoto maps, characteristic striations of dunes were also depicted, while in the inner part of the coastal zone, a sand ridge of the beach barrier can be partially traced. In particular, two beach barrier zones extend in the area, with the long one orientated NE–SW (Figure 3) and the short one NW–SE.

Through geomorphological mapping, it was observed that the west side of the plain is covered mostly by fluvial deposits of abandoned riverbeds and deltaic deposits. This fact is consistent with the geomorphological evolution of the area, since before 1945, the estuaries of the Nestos River were placed in the west part of the plain, where the anthropogenic factor had not gained so much influence. Today, in this area, the Kavala's International Airport (KVA) is situated, constructed in the early 1980s.

After 1952, when entrenchment and diversion of the river Nestos occurred [60], geomorphological modifications began. In order to illustrate those differences, we used declassified imagery KH-4 from 1960 and 1968. Most of the river channels and courses crossing the west part of delta were dried up and deprived this part of the river's delta of water and sediments. As a result, variation of the shoreline positions started with accretion at the mouth of the river and erosion of the coastal landforms [61].

In this work, the new shoreline zone at the mouth of the Nestos River was mapped in detail and the spatial distribution of deltaic deposits and dunes were delineated. Furthermore, anthropogenic levees, which diverted the river course axis to a direction almost N–S, and recent river formations were identified and mapped. Finally, darker areas of salt-marsh deposits were added in the broader areas of lagoonal formations and between deltaic deposits (Figures 3 and 4). In addition, satellite imagery (Copernicus Sentinel-2 and very high resolution images from Google Earth) were used in order to observe the current state of the floodplain.

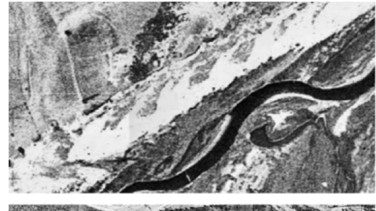

Beach Barrier
Narrow and elongate sand ridge situated parallel to the shoreline. It divides the mainland and the nearshore wetland. Two beach barrier zones extend in the area, with the long one orientated ENE-WSW and the short one NW-SE.

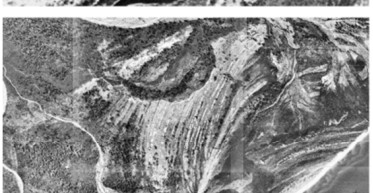

Coastal dunes
Aeolian landforms, which are formed by the wind and protect the back-barrier environments against wind and waves. In Nestos Delta, dunes are extended on the coastal zone, on ridges, spits and between deltaic and salt-marsh deposits.

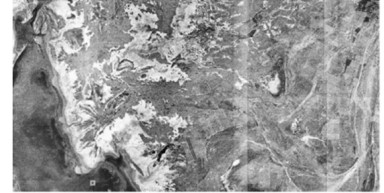

Deltaic deposits
Expanded sections of clastic sedimentary facies, result from depositional processes of Nestos river. They are forming mostly on the west side of the plain; due to the old estuaries of Nestos; and on current delta.

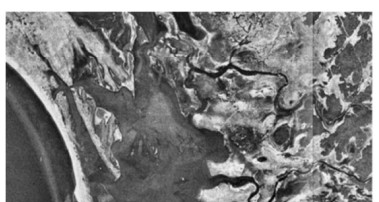

Lagoons
coastal bodies of water that have very limited connection to the open ocean. A lagoon system were formed on the west side of the plain.

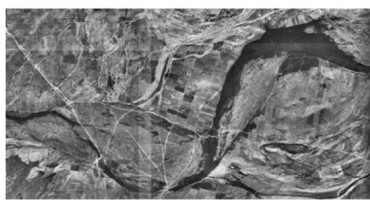

Point Bar
Features made of alluvial deposits that concentrate on the inside bend of meanders and river beds. In Nestos plain most of them are extending on the west side, were the old delta were formed.

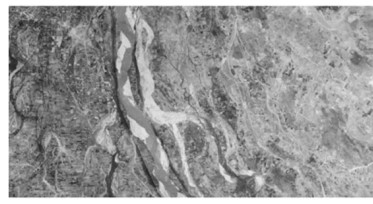

Current river bed
Current river course axis, after its diversion by the anthropogenic levee. The whitest channels are the older main paths and the darker the latest.

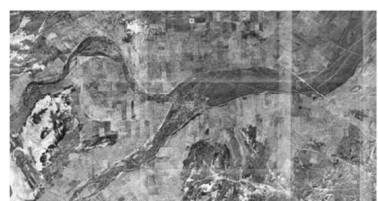

Old/abandoned meanders
River channels and courses crossing the west part of delta, which after the entrenchment and diversion of Nestos were dried up.

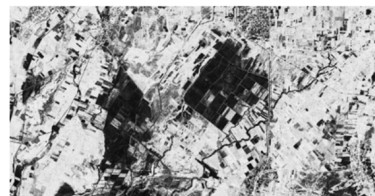

Marsh deposits
Extended network of marsh deposits and saturated soils is detected along the old and recent river beds. Also close to lagoons and coastal deposits as salt marshes.

**Figure 3.** Mapping of the geomorphological features using orthophoto maps from 1945 and KH-4 Corona frames.

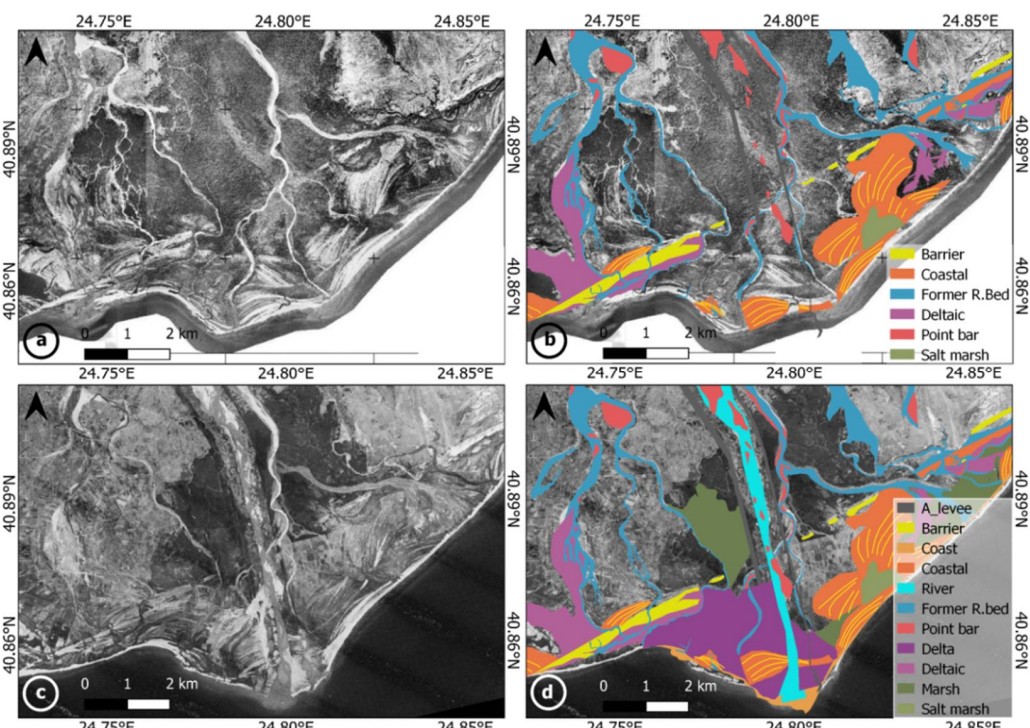

**Figure 4.** Geomorphological formations at the coastal zone of Nestos River delta. (**a**) orthophotomap from 1945, (**b**) mapping of old meanders, dunes and beach barrier in orthophoto map from 1945, (**c**) satellite image from 1960, (**d**) accretion of mouth river and extension of deltaic deposits and dunes.

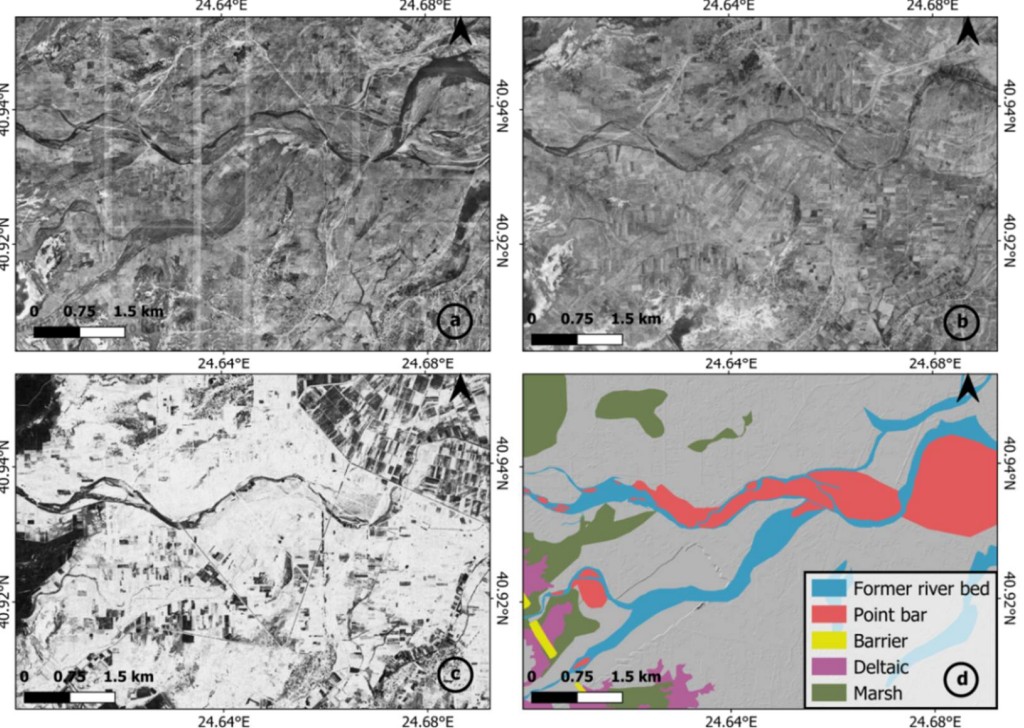

**Figure 5.** Recent evolution of meandering streams and progression of irrigation and drainage modifications according to (**a**) the orthophoto map from 1945 derived from Hellenic Cadastre, (**b**) KH-4 image from 1960 and (**c**) KH-4 image from 1968, (**d**) mapped geomorphological fluvial and deltaic formations.

Combining all these layers of information with the local topography (digital surface model of the area by Hellenic Cadastre with a 5 m ground resolution), the newly mapped geomorphological formations of the Nestos River plain were overlaid on the initially less-detailed geological map. In this geomorphological map that was compiled for the purposes of this study, 15 surficial geology units were presented, such as abandoned/old river channels, point-bars and marsh deposits. Along the coastal zone, aeolian/dune formations and deltaic deposits were identified and beach barriers were mapped. The internal part of the Nestos River channel was enriched with point-bars, in contrast with the external one where anthropogenic and natural levees were traced. Finally, lagoons and oxbow lakes in the floodplain were also mapped (Table 2, Figure 6).

**Table 2.** Geological units according to HSGME map and geomorphological units according to satellite and aerial imagery processing that took place for the purposes of this study.

| Geological Map by HSGME | Geomorphological Map Based on Aerial and Satellite Imagery |
|---|---|
| Holocene Floodplain deposits | Holocene Floodplain deposits |
| Holocene Nestos River | <500 yr Current River channel |
| Holocene Channel deposits | <500 yr Former River bed |
| | <500 yr Point-Bar |
| | <500 yr Anthropogenic Levee |
| | <500 yr Oxbow |
| | <500 yr Levee |
| | <500 yr Coastal deposits-dunes |
| Holocene Coastal deposits | <500 yr Coast |
| | Holocene Beach Barrier |
| | <500 yr Delta-deltaic deposits |
| Holocene Swamp deposits | Holocene Marsh deposits |
| | Holocene Salt marsh deposits |
| Holocene Lagoonal deposits | Holocene Lagoonal deposits |
| Pleistocene deposits (screes, tal) | Pleistocene deposits |

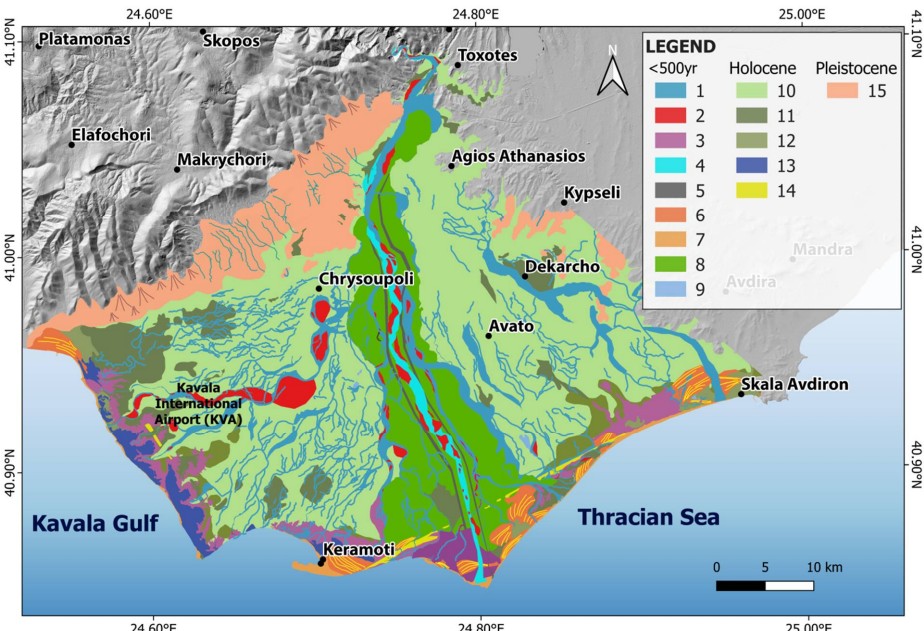

**Figure 6.** The new geomorphological map of the Nestos River floodplain resulting from the combination of orthophoto maps (1945) and Corona KH-4 images (1960, 1968). (1) former riverbed, (2) point-bar, (3) deltaic deposits, (4) current river channel, (5) anthropogenic levee, (6) coastal deposits (dunes), (7) coast, (8) levee, (9) oxbow, (10) floodplain deposits, (11) marsh deposits, (12) saltmarsh deposits, (13) lagoon, (14) beach barrier, (15) Pleistocene deposits.

The criteria that we used to assess the possible age of the sediments were based on expert judgment correlating information provided by HSGME geological maps, the relative surface modifications and the topographic position of the units. Afterwards, surficial geological deposits were classified in three categories in order to correspond with the applied methodology of Youd and Perkins:

1. <500 yr: Consists of fluvial deposits forming abandoned stream/meanders, point-bars, deltas and levees and also coastal deposits and oxbow lakes.
2. Holocene: In this category, the lagoonal and marshy deposits are grouped, not only surrounding the lagoons (saltmarsh) but also developed in the most internal parts of the floodplain. Consequently, the geomorphological features of the old beach barriers were included.
3. Pleistocene: Deposits of alternating coarse (sands and gravels) and fine (clays, silts) grained materials and screes.

## 3. Results

Using data provided both by the available geological maps of HSGME and the new geomorphological map, we were able to define the age and depositional environment of geological units. Having assessed this information, it was feasible to apply the approach suggested by Youd and Perkins criteria [25] in order to classify the geological units according to their susceptibility to liquefaction.

Initially, using as a reference the information provided by the official geological maps compiled by HSGME, only eight geological formations are depicted, classified as floodplain, coastal, swamp, lagoonal, and Nestos River deposits of the Holocene era and screes in alternation with deposits derived from fluvial processes and erosion in the upper limits of the plain from the Pleistocene era. Applying this approach on these maps, three classes of susceptibility to liquefaction units were defined; low, moderate and high (Table 3, Figure 7), while the non-susceptible to liquefaction Pleistocene deposits formed a separate class. In particular, the geological units were classified as:

- Low susceptibility: swamp deposits
- Moderate susceptibility: floodplain, coastal deposits, lagoonal
- High susceptibility: current river, channel deposits
- Non liquefiable: Pleistocene deposits, screes

The fluvial sediments, consisting only of channel deposits that are described as Holocene alluvial deposits according to HSGME, were classified as high susceptibility units, without being able to consider the location and the different behavior of the abandoned meanders or channels. The coastal deposits units were classified as moderate, while it is critical to point out that no delineation of dunes, beach barrier formation or the estuary deposits of the Nestos River are provided. This lack of information regarding both the discrimination of the type of surficial geological units and their age leads to a lesser accuracy of the assessment of liquefaction susceptibility and consequently to a coarser compilation of the relevant map.

On the other hand, taking into account the spatial distribution of deposits, as it is shown on the geomorphological map of the area proposed by this study, four different liquefaction-related classes were defined, plus the non-liquefiable one of cohesive Pleistocene deposits (Figure 8, Table 4). Having delineated 7 new formations on the geomorphological map, the total number of deposits that were examined regarding their liquefaction susceptibility was 15. Furthermore, deposits from the Holocene era were sub-grouped based on their age as (a) deposits younger than 500 years and (b) Holocene age deposits.

Considering as a base layer of information this detailed map e.g. geomorphological one, it was feasible to accurately assess the liquefaction susceptibility of the deposits on a regional scale and compile a reliable relevant map. This map is resulted based on the following classification regarding the degree of liquefaction susceptibility:

- Low susceptibility: swamp and marsh deposits
- Moderate susceptibility: natural levees, floodplain deposits, beach barriers, lagoonal deposits
- High susceptibility: coastal deposits, coast, oxbow lakes
- Very high susceptibility: former riverbed, point-bars, deltaic deposits, current river channel, anthropogenic levee.

Comparing the two liquefaction susceptibility maps, it is evident that differences exist between them. In particular, having mapped the additional geomorphological units, we observed that the zones of high to very high susceptibility are now extending across the whole floodplain and not only in the current main channel area of the Nestos River. Fluvial deposits were discriminated further to abandoned river channels, point-bars, levees and oxbow formations of the last 500 years and, consequently, were classified as high to very high susceptible units. Along the coast, beach barriers, dunes, coastal and deltaic formations of Nestos estuaries were also included, upgrading the coastal zone from moderate to high.

**Table 3.** Liquefaction susceptibility classes of the Nestos River delta, according to the geological map provided by HSGME.

| Geological Units Description | Classification of Liquefaction Susceptibility [25] |
|---|---|
| Holocene | |
| Floodplain deposits | Moderate |
| Channel deposits | High |
| Coastal deposits | Moderate |
| Swamp deposits | Low |
| Nestos River | High |
| Lagoons | Moderate |
| **Pleiocene–Pleistocene** | |
| Pleistocene deposits | Non liquefied |
| Scree | Non liquefied |

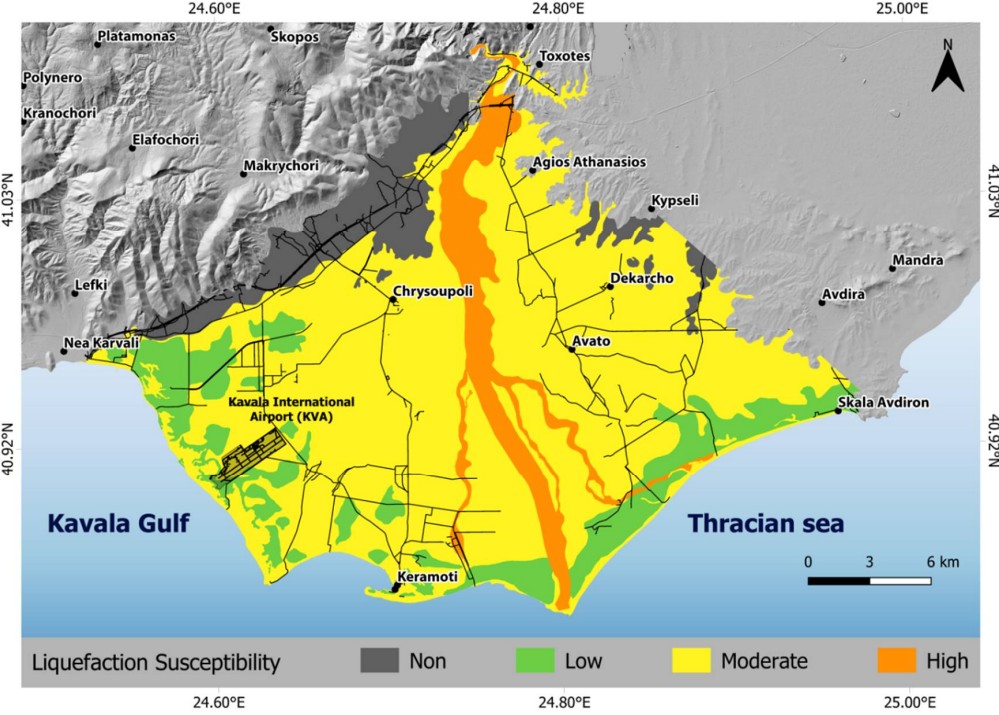

**Figure 7.** Liquefaction susceptibility map of the Nestos River delta derived from the application of the criteria proposed by Youd and Perkins [25] to the geological maps provided by HSGME.

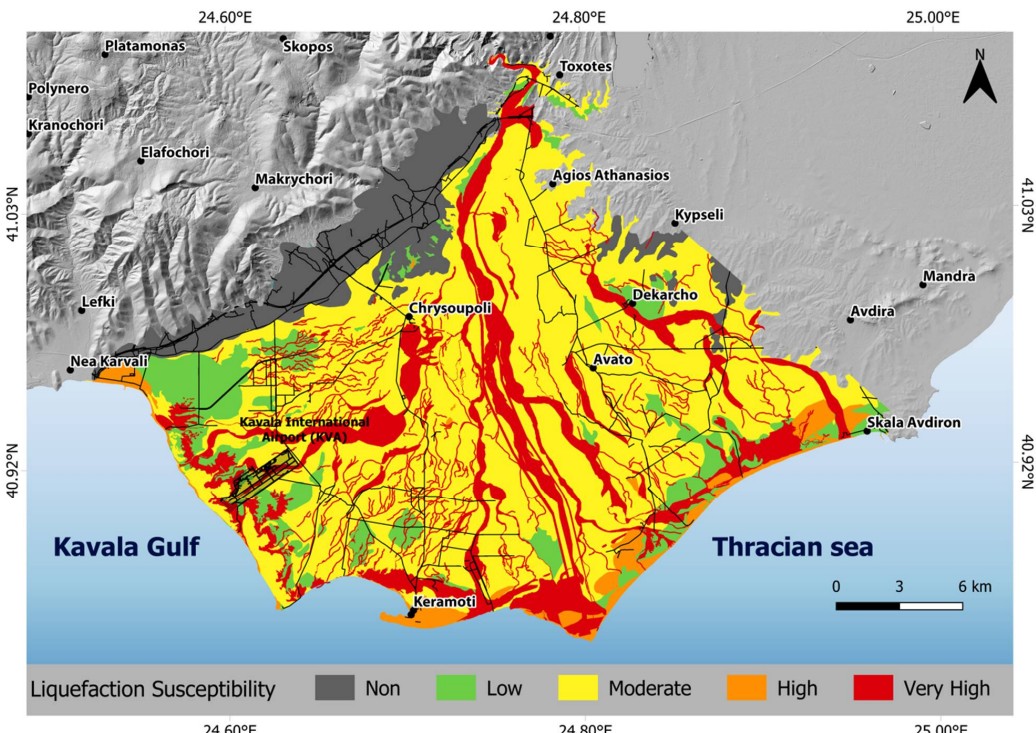

**Figure 8.** Liquefaction susceptibility map of the Nestos River delta, resulting from the application of Youd and Perkins criteria [25] to the new geomorphological map, produced by processing of satellite and aerial imagery.

**Table 4.** Liquefaction susceptibility classes of the Nestos River delta according to the geomorphological map produced by processing of satellite and aerial imagery.

| Geological Units Description | Classification of Liquefaction Susceptibility [25] |
| --- | --- |
| <500 yr | |
| Former riverbed | Very High |
| Point-bar | Very High |
| Deltaic deposits | Very High |
| Current river channel | Very High |
| Anthropogenic levee | Very High |
| Coastal deposits (Dunes) | High |
| Coast | High |
| Levee | Moderate |
| Oxbow | High |
| **Holocene** | |
| Floodplain | Moderate |
| Marsh | Low |
| Saltmarsh | Low |
| Lagoonal | Moderate |
| Beach barrier | Moderate |
| **Pleistocene** | |
| Screes, deposits | Non liquefied |

According to this detailed liquefaction susceptibility map, 85.56 km$^2$ (16.66%) (Table 5) of the study area classified as a very high susceptibility zone, covered mostly by fluvial deposits. The presence of these deposits is more extensive in the western part of the plain due to the recently abandoned estuaries of Nestos, which were dried after entrenchment and diversion of the river. After those modifications in the delta plain land, the Kavala International Airport (KVA) was constructed in the early 1980s. In particular, using as a base layer the geological map of HSGME and focusing on this critical infrastructure, we observe that KVA is founded over the floodplain and swamp deposits of Nestos (Figure 9a).

However, according to the new geomorphological map, the airport was constructed over the old/abandoned Nestos River channels. Consequently, in the former case, the airport's area is classified as low to moderate liquefaction susceptibility, while based on the latter scenario, as high to very high susceptibility to liquefaction (Figure 9b).

**Table 5.** Liquefaction susceptibility classes areas in the Nestos River floodplain, according to the new liquefaction susceptibility map.

| Liquefaction Susceptibility | Susceptibility Area (km$^2$) | Susceptibility Area (%) |
|---|---|---|
| Non-liquefiable | 58.40 | 11.37 |
| Low | 51.00 | 9.93 |
| Moderate | 299.76 | 58.36 |
| High | 18.88 | 3.68 |
| Very High | 85.56 | 16.66 |
| **SUM** | **513.60** | **100.00** |

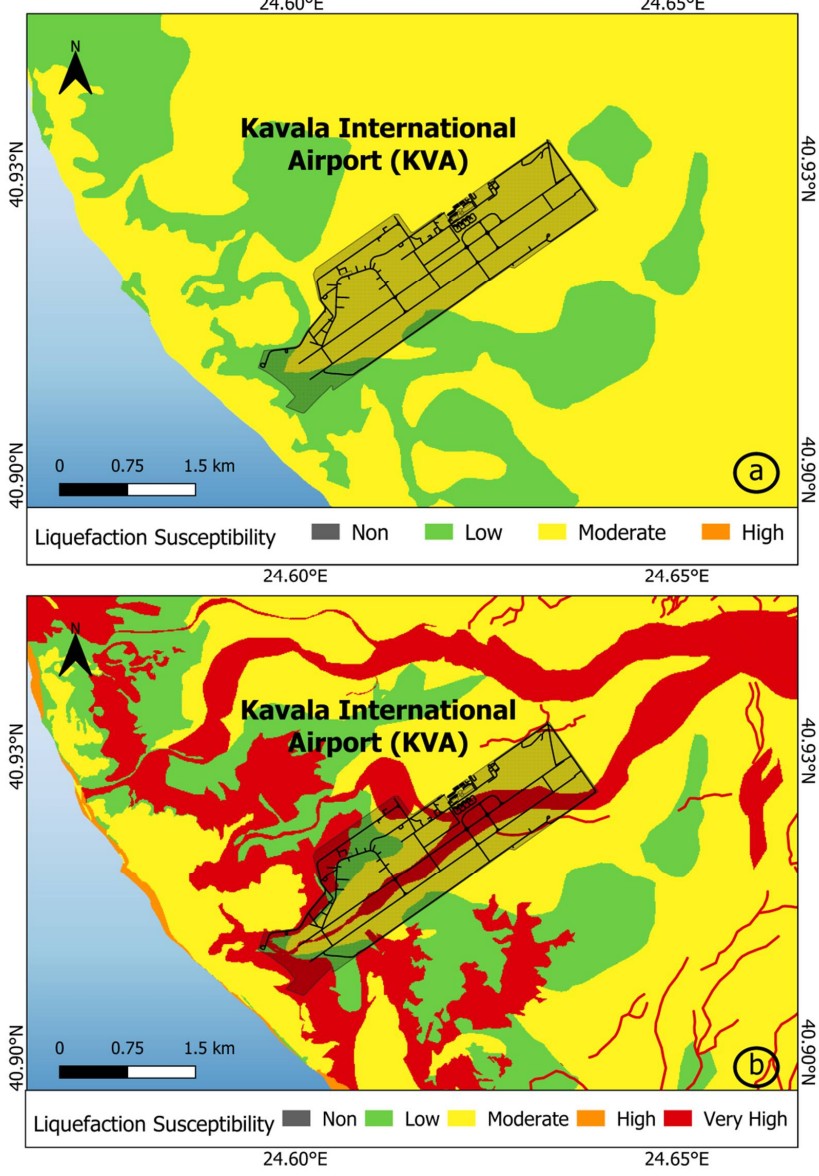

**Figure 9.** Comparison of liquefaction susceptibility maps focusing on the critical infrastructure of Kavala International Airport (KVA) (**a**) based on the geological map of HSGME, (**b**) based on the new geomorphological map produced by processing of satellite and aerial imagery.

## 4. Conclusions

In the current study, the liquefaction susceptibility of geological units at the Nestos River delta (Greece) was assessed. The motivation for this research was: (a) the presence of tectonic structures that are capable of generating large earthquakes in the broader area and the high potential onshore faults, (b) the existence of recent and late Holocene sediments and (c) the critical infrastructures that were designed and constructed within the study area over the last 40 years.

The crucial information of the depositional environment of soil units is usually provided by a geomorphological map. However, in our case there was a lack of detailed information regarding both the type of deposits and their spatial distribution. In order to overcome this, we used satellite and aerial imagery aiming to compile a geomorphological-oriented map, instead of a simplified geological one, that would help us to screen the river deposits. In particular, using remote sensing data, we were able to delineate fluvial and coastal geomorphological features such as floodplains, estuaries and the locations of abandoned stream/meanders and oxbow lakes. Afterwards, these features were additionally grouped into three categories based on their age: younger than 500 yr, Holocene and Pleistocene deposits. To assess the liquefaction susceptibility, we applied the criteria suggested by [25], according to which the recently deposited sediments were classified as high–very high susceptibility and pre-Pleistocene deposits as non-liquefiable ones.

Thus, a liquefaction susceptibility map was developed, classifying the Nestos River delta in four susceptibility classes: low, moderate, high and very high. The most prone to liquefaction class covers the 16.66% (85.56 km$^2$) of the study area and consists of coastal deposits of dunes, beach barriers and deltaic formations and fluvial deposits of old/abandoned meanders, current river channels and point-bars.

Using the liquefaction susceptibility map as a guide to delineate the most prone to liquefaction areas led us to some important conclusions regarding the location of one specific critical infrastructure. In particular, it is shown that the International Kavala Airport was constructed on the old estuaries of the Nestos River on the west side of the plain, an area that is classified as having high–very high susceptibility for liquefaction. Thus, it is emphasized the importance of the detailed geomorphological mapping of soil units for accurately assessing the liquefaction susceptibility. In addition, this case highlighted the necessity of conducting a detailed geotechnical investigation based on in-situ tests at the area of the airport for evaluating the liquefaction potential of soil units and the relevant induced displacements.

**Supplementary Materials:** The following supporting information can be downloaded at: https://www.mdpi.com/article/10.3390/geosciences12100361/s1, Figure S1: Outline of KH-4 Corona frames and ground control points used for orthorectification [49,55]; Figure S2: (**a**) Piezometric map of the phreatic/unconfined aquifer in October 1978. Piezometric isolines with thick blue lines (1 m interval). (**b**) Topographic map of Nestos delta area, with 1 m contours. Contours extracted and simplified from FABDEM digital terrain model; Table S1: KH-4 Corona declassified imagery frames used; Table S2: Ground control point coordinates (meters) and residuals after triangulation with KH-4A frame DS009009009DV081; Table S3: Ground control point coordinates (meters) and residuals after triangulation with KH-4B frame DS1104-1058DF092.

**Author Contributions:** Conceptualization, S.V. and G.P.; Methodology, M.T. and S.V.; Investigation, M.T. and S.V.; Data curation, S.V. and G.P.; Writing—original draft, M.T. and S.V.; Writing—review & editing, G.P.; Supervision, G.P.; Project administration, N.K. and I.D. All authors have read and agreed to the published version of the manuscript.

**Funding:** We acknowledge support of this work by the project "Risk and Resilience Assessment Center–Prefecture of East Macedonia and Thrace-Greece". (MIS 5047293) which is implemented under the Action "Reinforcement of the Research and Innovation Infrastructure", funded by the Operational Programme "Competitiveness, Entrepreneurship and Innovation" (NSRF 2014–2020) and co-financed by Greece and the European Union (European Regional Development Fund).

**Institutional Review Board Statement:** Not applicable.

**Informed Consent Statement:** Not applicable.

**Data Availability Statement:** Not applicable.

**Conflicts of Interest:** The authors declare no conflict of interest.

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
