# Peer review of "A Detailed Liquefaction Susceptibility Map of Nestos River Delta, Thrace, Greece Based on Surficial Geology and Geomorphology"

_geosciences, doi:10.3390/geosciences12100361_

Round 1

Reviewer 1 Report

Geosciences article "A detailed liquefaction susceptibility map of Nestos river delta, Thrace Greece based on sufficient geology and geomorphology."

Reviewer recommendation: Reconsider after major revision, with encouragement to resubmit

The authors are commended for using optical imagery and geomorphology to classify geologic units for the purposes of mapping liquefaction susceptibility.  They have done an excellent job of applying their geologic skills and identifying unit types.

Unfortunately, there is more to liquefaction susceptibility than mapping and identifying geologic units.  The textural characteristics of soil and porosity (specifically, relative density between the loosest and densest possible states) are critically important.  This critical aspect is defined commonly by the 'Chinese criteria' for liquefaction susceptibility and has been more recently studied and redefined by Bray (Recent H.B. Seed lecture; Bray et al., 2004a,b; Bray and Sancio, 2006).  For example, the Nesto's River is identified as having very high liquefaction susceptibility.  Still, there are no grain size data nor characterization to determine whether the main stem and distributaries of the river have high plasticity clay components that would eliminate liquefaction hazards.  

The relative density of the soil also controls liquefaction susceptibility.  A variety of techniques have been used as proxy measurements of relative density in the field.  These include standard penetration test, cone penetration test, dilatometer penetration test, Becker hammer test, and Shear wave velocity.  The reviewer was anticipating that once the geologic units have been defined, the authors would pursue an analysis of geotechnical logs to determine the distribution of penetration resistances or velocity used for each geologic unit.

Together, aerial mapping of geologic units and geomorphology, the characterization of fines content within each geologic unit, and the analysis of penetration characteristics or shear wave velocity for each geologic unit would serve as the backbone of a credible liquefaction susceptibility map.  If a proxy method is used to map susceptibility, it must be ground truth or calibrated using these basic elements.  The authors have done an excellent job mapping specific geologic units, but use non-specific literature criteria to map a small area as having high and extensive liquefaction susceptibility.  The authors may be accurate in the assessment but that would be entirely by chance.  No effort has gone into calibrating the geologic units with real soil data that would indicate textural or relative density susceptibility.  The authors may have created an alarming map that over-predicts susceptibility.  Only ground truth studies can determine the effectiveness of the effort.

The authors can easily and quickly remedy this problem.  The reviewer recommends that they collaborate with a local engineering geologist or civil engineer to classify each of the geologic unit types from existing borehole data.  For a rough non-site specific classification this could be done quickly.  Textual information, as well as statistical distributions of penetration resistance or velocity per unit, would quickly greatly improve with mapping effort.  The authors are advised to read the U.S. National Research Council report: State of the art and practice in assessment of earthquake – induced soil liquefaction and its consequences.  All of these topics and issues are discussed in detail, therein. 

https://www.nationalacademies.org/our-work/state-of-the-art-and-practice-in-earthquake-induced-soil-liquefaction-assessment 

With a ground truth calibration, the authors paper would be greatly improved and likely be accepted  The reviewer encourages the authors to add these components and resubmit.

Reviewer 2 Report

I liked the manuscript submitted and the methodology applied, the approach proposed is well presented and the results are of interest for the region and in general for the scientific community.

General comments:

References: there are too many references related to conference abstract or proceedings (see ref 14-15-20-26-40-41-46-49-57) and some authors statements are solely based on this type of documents, I suggest to add more “strong” references also to give the reader a larger opportunity to read and verify the information suggested.

Names of geomorphological elements or river or tectonic structures or cities should be visible in the manuscript figures, please add all the names mentioned or delete them.

The question of the age of the surficial sediments described in the text and highlighted in the new map is a critical point, authors should mention that no ages from cores or direct dating from the authors has been used and the attribution of the ages is a tentative effort based on geomorphological consideration and previous geological mapping.

The road network discussion is not well presented and considering that in my opinion this part is not relevant as the airport infrastructure it can be deleted from the manuscript. Differently, if the authors want to include the road network some more effort is needed to properly present their point of view.

Detailed revision

Line 12: as for the Holocene subdivision I suggest to refer to http://quaternary.stratigraphy.org/iugs-ratifies-holocene-subdivision/

The recent Holocene time interval as suggested by the authors (less than 500 yr.) is an expert judgement and it should be clearly stated and clarified in the text.

Line 30: the road network mentioned is not shown in any figure and any consideration on its liquefaction susceptibility cannot be evaluated, I suggest to erase this part (it is of secondary relevance with respect to the main airport infrastructure) or to provide a map, a new figure possibly, where a comparison of the road network with the new geomorphological map is displayed.

Line 101-102-103: the presentation of the extensional regime and orientation of active faults at regional scale cannot be based on ref 40 which is related to the study of a single archeological site, there was probably an error (should it be ref 41?) and a strong reference is absolutely needed.

Line 103 to 111: as for the presentation of the active faults and their seismogenetic potential there Is a clear lack of references, this cannot be accepted, please refer at least to databases available at European scale (https://seismofaults.eu/) or specific national compilation like http://gredass.unife.it

Line 114: the Evros river is not visible in any figures?

Line 125: the question of the age of the surficial sediments described in the text and highlighted in the new map is a critical point, authors should mention that no ages from cores or direct datings from the authors has been used and the attribution of the ages is a tentative based on geomorphological consideration and previous geological mapping.

Line 129: ref 41 seems to be not appropriate, please verify.

Line 143: Evros delta not visible in any figure, please provide it in the map.

Line 149: the 45 to 48 references are 40 yrs. Old, a more recent study is welcome

Line 150 to 156: specific reference is needed, the rift valley of Nestos-Prinos is not shown in any figure

Line 163-164: soil materials from 2.5 to 6 km? I do not understand but soil thickness of several km is really difficult to imagine. Please revise this sentence.

Line 187-188: the 51 and 52 references are 40 yrs. old, how we may derive the actual water table depth range from these data?

Line 188 to 193: a map showing the boreholes location and thus their distribution is needed (could be added to figure2?), a reference to the 2010-2012 results is also more than welcome. I suggest to provide a water depth range based on the data above.

Line 194 to 197: see comments provided for Line 125.

Line 199 to 207: please rewrite this section in order to make it clearer, it was difficult to follow and also please correct the age in fig. 2 where units 5 to 8 are Pleistocene not in agreement with the text.

Line 253: should the authors add Figure 4?

Line 260 to 262: the orientation of the two main beach barrier zones is not correct, probably, from figure 4 and 6 the long one is NE-SW striking, isn’t it?

it is difficult to use figure 3 for references since the Figure 3 has several problems: images are not oriented, lack of differentiation between ortho and Corona images, lack of scale.

A general map showing the location of the small frames is also welcome.

Line 269: a reference for the 1952 works performed on the river is needed.

Line 275: please delete “Thus” and insert more clearly “In this work”

Line 293: ..to assess the POSSIBLE age….

Line 297: again the question of the age, sincerely I do not like the authors definition but please consider that in the literature Late Upper Holocene means younger than 4.2 Ky, see also  “Formal subdivision of the Holocene Series/Epoch: a Discussion Paper by a Working Group of INTIMATE (Integration of ice-core, marine and terrestrial records) and the Subcommission on Quaternary Stratigraphy (International Commission on Stratigraphy) by Walker et al., 2012, JOURNAL OF QUATERNARY SCIENCE (2012) 27(7) 649–659” and consider previous comments as Line 12: as for the Holocene subdivision I suggest to refer to http://quaternary.stratigraphy.org/iugs-ratifies-holocene-subdivision/

Line 327: ..”define in detail “ is not appropriate since no direct age data can be provided, truly speaking the authors are “suggesting POSSIBLE age based on …”

Line 332-333: this part is a bit confusing, the HSGME map of figure presents some Pleistocene deposits and scree of undefined age, I suggest to rephrase in a clearer way.

Line 359-360: please consider the suggestion provided for Line 12, the very recent Holocene time interval as suggested by the authors (less than 500 yr.) is an expert judgement and it should be clearly stated and clarified in the text. For example, why and on which bases the 500 yrs. age has been used? Why not 1000 yrs.?

Line 386: upper Holocene? 500 yrs. Or younger than 4.2 Ky?

Line 402 to 408: the road network described in the text is not presented in any figure, before discussing any implication regarding these type of infrastrucuters the authors should present a figure with the road network plotted over the new geomorphological/liquefaction map.

I am not sure that this point may deserve such a discussion but it is up to the authors a decision on it.

Abstract and conclusion are fine but should be revised by the authors after the necessary changes need for the text.

Round 2

Reviewer 1 Report

Title: A detailed liquefaction susceptibility map of Nestos River delta, Thrace Greece based on surficial geology and geomorphology.     Unfortunately, the authors did not respond to the reviewer comments in a meaningful way.  There is an airport that almost certainly has subsurface textural data that has not been investigated.  likewise, numerous small communities with wells and therefore soil boring were not pursued.  The authors are incorrect that CPT and the parameter Ic cannot be used as a mapping research tool to assess soil texture and liquefaction potential.   The authors refer to guidelines in the US NRC document on liquefaction pages 111-112.  I have both the 1985 and 2021 documents in front of me, and there is nothing on these pages even topical to the authors' points.  The Youd and Perkins method is five decade old and in the absence of subsurface data is not a credible "stand-alone" method for mapping in 2022.  Reviewer recommends declining the paper. 

Reviewer 2 Report

Dear Authors

thanks for your effort done in order to address all my comments, I really appreciate reading the second version of the manuscript that has been implemented and improved at a good level.

Few notes:

Materials and methods: the suggestions made for the estimation of the age of the sediments and the level of the aquifer were followed and properly addressed adding figure S2. At line 306 there is a strange message of error.

Results: I do not understand why table 3 (lines 425-426) with road network classification based on potential liquefaction is still here, shouldn’t be erased as confirmed in the response letter by the authors? There are two table 3 by maintaining this and I suggest to erase it. It is probably a mistake done during the revision.

All the best

Author Response

Dear reviewer,

We would like to thank you for your comments. As it can be seen in the revised manuscript, the relevant column of road network in Table 5 is erased following the suggestion.

with kind regards